# Monitoring of Surgical Wounds with Purely Textile, Measuring Wound Pads—III: Detection of Bleeding or Seroma Discharge by the Measurement of Wound Weeping

Harald Pötzschke and Kai Zirk *

Faculty of Electrical Engineering/Mechatronics, PHWT—Private University for Economics and Engineering, D-49356 Diepholz, Germany
* Correspondence: zirk@phwt.de

**Abstract:** To enable stating a final common sensor design of purely textile, measuring wound pads for the monitoring of surgically provided wounds with regard to tissue temperature, moisture release and stretching (as indicators for the most prominent wound healing disruptions bacterial inflammation, bleeding/seroma discharge, and haematoma/seroma formation), the aim of this investigation was to identify and quantify possible variables practically affecting the detection of water in a systematic study. The textile sensors comprise insulated electrical wires stitched onto a textile backing and parallel wires form a plane sensor structure whose electrical capacitance is increased by water (contained in blood or lymph) in the textiles. Only parallel sensor wires forming double meanders were examined because this structure enables all the parameters of interest to be measured. Surprisingly the results are complex, neither simple nor consistent. The change in electrical capacitance (measuring signal) upon the standardized addition of water was not additive, i.e., it was not found to be correlated to the moistened area of the sensor array, but inversely correlated to the diameter of the sensor wire, mildly pronounced in connection with smaller stitching spacing (stitching loops along the sensor wires). The measuring signal reached a maximum with medium sensor wire spacings and pronounced with a smaller stitching spacing. Without exception, the measuring signal was systematically higher in connection with smaller (compared with larger) stitching spacings. The results presented indicate that the optimization of the capacitive textile sensors cannot be calculated but must instead be carried out empirically.

**Keywords:** wound monitoring; full textile wound pads; bleeding; seroma fluid discharge; wound wetness; electrical capacitance

## 1. Introduction

For surgical wounds (clean incisions closed with staples or sutures) for planned healing by first intention (*sanatio per primam intentionem*/p.p., without the prior regrowth of missing tissue, namely, healing immediately), protecting the wound from the environment is the most important, primary purpose of a dressing [1–3]. However, furthermore, a wound dressing may be a preferential location for sensors for monitoring the actual common condition of the wound.

Novel textile sensors (as part of a wound dressing-based measuring system) in development used to determine characteristics indicating the condition of acute, medically treated wounds consist of very thin, insulated electrical conductors (sensor wires) stitched onto textile backing. Such textile sensors can be integrated into both textile and several types of non-textile wound dressings. When used with a measuring and assistance system, sensor textiles enable wounds to be monitored by tracking the course of certain wound properties [4–7]. Using a smart dressing to detect changes in the moisture (liquid water) above the wound can provide an indication of bleeding or the discharge of intercellular fluid (such as lymph from a postoperative seroma) [4].





Two parallel, electrically insulated wires are stitched onto a textile backing to form a planar (two-dimensional) array of electrical conductors. This sensor structure behaves similarly to an electrical capacitor. Liquid water flowing or diffusing in the near field of the textile (as a new dielectric) increases the dielectric conductivity (relative permittivity $\varepsilon_r$) and thus the electrical capacitance of the array of conductors. The difference between relative permittivities is large (for example, the relative permittivity of pure water is about 80-fold that of pure air), whereas the temperature correlation for water $\Delta\varepsilon_{r,H2O}/\Delta T$ is only an estimated—0.4 °C in the temperature range considered [8].

However, the electrical capacitances are almost impossible to estimate, let alone calculate. The field lines of the electrical fields between oppositely charged conductors span the entire space, and only a small part crosses the section of textile becoming moist. Furthermore, the local degree of moisture—which may range from merely the surfaces of the yarns making up the textile becoming damp to all the cavities of the textile being filled with moisture—is unknown. To measure an electrical capacitance with textile sensors, parallel, electrically separated, insulated sensor wires are used. Since substantial, measurable changes are required, the conductor array should consist of sections of sensor wire that are as long as possible and arranged in parallel [4].

Surgically supplied wounds are predominantly covered using traditional textile dressings since 'modern' (moist, so-called 'smart' or reactive) dressings in principle offer no advantage but are much costlier. Textiles in the form of woven, knitted, braided and non-woven fabrics are used in direct contact with a postoperative wound. Other than synthetic fibres (PES, PP, PE), dressings are typically composed mostly of biocompatible polysaccharides: cotton and viscose fibres in the form of, say, traditionally woven gauze compresses. These are very soft and supple, and their absorbency and air permeability are defining characteristics. Their tendency to stick to the wound on direct contact can be effectively reduced by using coatings.

Adverse events may disturb (and slow down) or even disrupt the primary healing of surgical wounds, which usually takes about 10 to 14 days [2]. Such adverse events primarily include:

(1)    initially localized increase in temperature and swelling due to bacterial inflammation [5],

(2)    haemorrhage or fluid discharge, with or without tearing of the suture (or staple line), and

(3)    swelling caused by haemorrhage into tissue or accumulation of fluid in *peri-vulneral* tissue [4].

To observe the signs of adverse events as early as possible, the condition of the wound is regularly examined by visual inspection, during which dressings are changed, despite it generally being beneficial that dressings are left untouched for a longer period of time—ideally until the removal of the stitches or staples.

Relevant, objective quantitative information about the condition of a wound can potentially be provided by a number of physical and chemical wound parameters (e.g., temperature, moisture, pH, oxygen partial pressure, *peri-vulneral* blood flow, etc.) [9]. These variables could be gauged by using wound dressings with measuring capability, in order to monitor the wound. There are various conceptual ways of integrating conventional sensors into wound dressings; however, they all have a medical drawback in common, namely that the properties of dressings become impaired. Sensors (which often—even as flexible foils—are more or less hard objects) positioned on a substrate or packed inside housing over a relatively large area either deteriorate the smoothness of the surface, the softness and the internal stiffness of dressings or only gather readings from small parts of the wound area.

If the wound dressings used are textile dressings, they need to be functionalized such that selected variables can be recorded continuously or with sufficient frequency. 'Smart textiles' work on the principle of very thin, electrically conductive yarn flexibly integrated into or onto a textile by means of weaving, embroidery, sewing, warp knitting or weft

knitting. The great advantage of smart, functionalized textiles, when used as dressings, is that they can still be soft and supple. There is no need for conventional sensors—hard, bulky, foreign bodies that might lead to pressure, friction or skin intolerance, etc., in or around the wound and could therefore cause further, highly undesirable injuries. Furthermore, the production of textile sensor structures is easy and relatively inexpensive in almost any size. Additionally, textile sensors provide a decisive advantage over traditional (specific) sensors that they—when made with an appropriate design—are capable of gauging multiple physical variables simultaneously.

The adequate monitoring of the healing of surgical wounds with functionalized textile dressings should detect signs of the above-mentioned most frequent disturbances of wound healing and therefore requires at least the *peri-vulneral* detection of any:

(1)   increase in temperature, indicating inflammation;
(2)   abrupt increase in wetness, indicating bleeding or wound seroma discharge; and possibly
(3)   (lateral) extension, indicating a volume increase due to tissue inflammation, haemorrhage or seroma formation. Textile sensors are capable of measuring all these variables [4–7]:

The theory of a possible textile-based capacitive moisture determination is described at length in an earlier publication [4]. The determination of moisture with a textile sensor is possible with sensor conductors that are electrically separated, insulated and arranged in parallel to each other. To obtain the highest possible measurable changes in capacitance, the arrangement of the sensor conductors should include sections which are as long as possible and in parallel to each other by creating the largest attainable number of double meander loops, for example. We found no information in the literature on the amount of fluid which is secreted by a wound healing irregularly and disturbed. Due to the enormous measurement effects to be expected, it should be capable of detecting the penetration of moisture into the carrier textile between the two sensor yarns over a section measuring 0.1% (amount arbitrarily determined) of the overall length of the sensor wire covering the wound area. This corresponds to a liquid quantity of only a few microlitres, an amount presumably below any clinical relevance.

The challenge of this investigation was to systematically identify and quantify the most relevant possible practical variables affecting the detection of water with purely textile, measuring wound pads for the monitoring of surgically provided wounds. Lastly, one combined sensor design with regard to the detection of tissue temperature, moisture and stretching (as indicators for the most prominent wound healing disruptions bacterial inflammation, bleeding/seroma discharge and haematoma/seroma formation) is objective and should be achieved.

For this purpose, purely textile, measuring wound pads were fabricated in a small pilot plant scale production process. The systematic investigation using these wound pads practically simulates the clinical situation only but does not focus on any theoretical considerations of physics and/or (expectable) quantity interrelation.

Textiles are traditionally and will be prospectively popular materials for dressings for acute wounds [1,3,10–12]. Sensing wound pads (wound dressings with integrated sensors) are an object of research with numerous published results [13–31]. However, there are usually major distinct differences in our research and development:

1.   The object of sensing wound pads (developed by others) are usually chronic wounds, i.e., ulcerations of the skin caused by necrosis. Probes detect in direct contact with the wound exudate covering the wound [13–31]. The reason for this conceptual preference for chronic wounds may be the fact that these wounds show a persistent wound healing disorder over a long period of time (weeks or even months) and should therefore 'obviously' benefit significantly from wound monitoring.
2.   Furthermore, many developments simultaneously try to integrate a controlled delivery of medicines from a wound pad material-based repository to promote wound healing.

## 2. Materials and Methods

### 2.1. Determining the Moisture

The technical skin model for the capacitive determination of the amount of water in wound dressings consisted of a small tub (internal dimensions: 170 mm · 90 mm · 50 mm, a volume of about 765 mL) covered with a piece of stretched plastic film (fish pond film, thickness: 1 mm) made of EPDM (ethylene propylene diene monomer) rubber or PVC (polyvinyl chloride). Temperature-adjusted water (35.0 °C ± 0.01 °C) from a circulating thermostat ('F 32-HL', Julabo, Seelbach, Germany) flowed through the tub (volume flow: about 700 mL/min) [1]. Figure 1 shows diagrams and a photograph of the model. A Pt100 temperature probe with the tip placed immediately beneath the centre of the dressing samples positioned on the film was used to measure the temperature directly underneath the film (about 35.0 °C ± 0.03 °C).

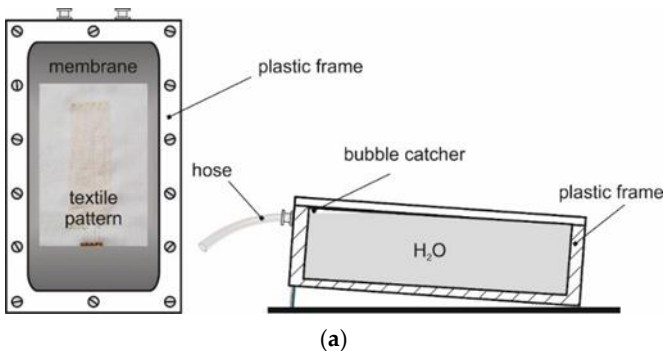
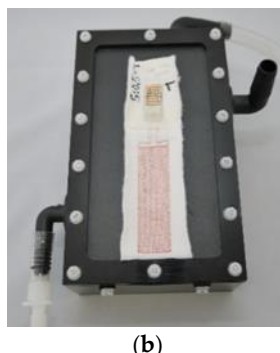

(**a**)      (**b**)

**Figure 1.** (**a**) Diagrams of the technical moisture model (top view and cross-section). (**b**) Photograph of the model.

Water was added to the wound pads by gentle mechanical pipetting with a piston-stroke pipette (Eppendorf pipette, 'Series 2000 Reference, variable 0.5–10 µL', Eppendorf, Hamburg, Germany). In accordance with DIN 12650, the inaccuracy of this method for a pipetted volume of 7.5 µL is estimated to be ± 1.55%. This estimate of the inaccuracy was derived by using linear interpolation based on the figure of ± 1.9% for 5 µL (inaccuracy of the mean (incorrectness) ± 1.5%, imprecision ≤ 0.8%) and ± 1.2% for 10 µL (inaccuracy of the mean ± 1.0%, imprecision ≤ 0.4%).

To determine the moisture, the dressing samples were placed centrally on the lower part of the surface film of the moisture model [4] and their electrical connections were hooked up to a precision impedance analyser. Readings were taken a few minutes later once a constant temperature had been reached, as confirmed by constant electrical capacitance. After the initial values had been recorded, a drop of water (7.5 µL) was added, and the values were recorded every 30 s until the results were constant (after 2 or 3 min). During this time, the water first flowed (or seeped) into the dressing textile and was then observed to spread out inside it before subsequently evaporating quickly, causing the measurements to decrease again within a few minutes.

Each dressing sample was measured three times. In between, the samples were dried in room air for at least 20 min (usually 30–45 min) until they were completely dry. This was confirmed by the electrical capacitance more or less returning to its original value before initial moistening.

## 2.2. Measuring Instruments

The electrical capacitances of the dressing samples were measured with a 'WK 6500/5 B Precision Impedance Analyser' (Wayne Kerr Electronics, Chichester, UK) with gold-plated terminals in four-wire technology. It was calibrated weekly with electronic precision capacitors, and its accuracy was checked daily. The measurements were read from the continuously changing values displayed by subjectively averaging the figures (which were updated approximately every second) to four significant digits.

To measure capacitance, the connections (at the same end) of two sensor wires positioned alongside each other (in particular connections 1 and 2, see Figure 2) were connected.

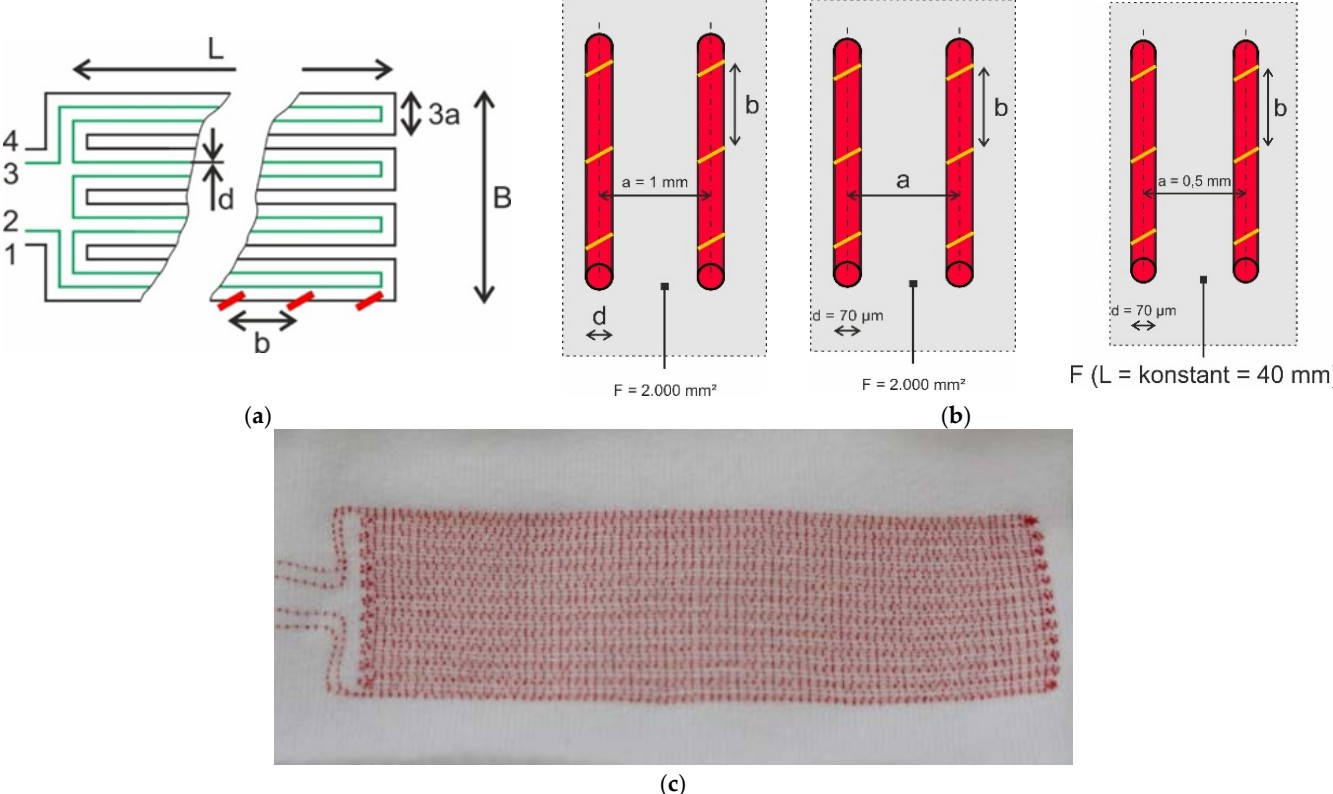

(**a**)                  (**b**)

(**c**)

**Figure 2.** (**a**): Symbolic diagram of the sensor wire array stitched onto the dressing samples. Of the connections on the left (1,2,3,4), 1 and 4 are connections at both ends of the same sensor wire, the same applies to connections 2 and 3. (**b**): Representation of the variables (larger letters) of the series from left to right: Series D (wire diameter d and stitching spacing b); Series S (conductor spacing a and stitching spacing b); Series Fx (sensor width B and stitching spacing b); (**c**): Photo showing an example of the wound dressings used.

The inaccuracy of the measurements depends on many factors of influence (measuring rate, measurement frequency (sampling frequency), temperature, the terminals used, etc.), and is only specified by the manufacturer for standard conditions. Our own estimates of the relative error (relative inaccuracy: $\delta x/x$) under the measuring conditions used (connection to gold-plated terminals in 4-wire technology) are $(\delta x/x) \cdot 100\% < 3\%$.

Temperatures were measured using Pt100 sensors with 4-wire technology ('Pt100 1/10 DIN accuracy SE 012' measuring probes with a PC-based measuring system 'PT-104' data logger for Pt100 sensors with 'PicoLog' display and processing software, Pico Technology, provided by PSE Priggen Special Electronics, Steinfurt, Germany). The inaccuracy of the measuring system was $\pm 0.058\,^{\circ}C$ (inaccuracy of the measuring system $\pm 0.01\,^{\circ}C$, inaccuracy of the probes $\pm 0.048\,^{\circ}C$ (DIN EN 60751).

*2.3. Textile Sensors*

Following the development of a suitable production process, the textile sensors were contract-manufactured by the ITA Institute for Textile Technology at RWTH Aachen University.

Cotton jersey ('white/heavy/elastic, type SW 45542-5003', Scheffer & Wiggers, Nordhorn, Germany; 95% cotton, 5% elastane) with 30% reversible stretch was used to make the textile backing. The textile backing was clamped between two layers of water-soluble PVA paper (SolvyFabric, Gunold, Stockstadt/D) in order to make stitching possible. Lukewarm water was later used to dissolve the PVA paper, and it was completely washed out using multiple rinses of clean water.

Enamelled copper wires ('Cu-ETP 99.95%', Elektrisola Dr Gerd Schildbach, Reichshof, Germany) were used for the electrical conductors, with the following diameters: 0.071 mm (71 μm), 0.14 mm (140 μm) and 0.21 mm (210 μm). They served at least as a model for silver wires, which would possibly be necessary for future medical applications on humans.

The sensor wires were stitched on in double meanders. Other possible geometries (e.g., double spirals) are not likely to yield any fundamental advantages. Moreover, theoretically, all the physical variables of interest can be determined with double meanders, these being the temperature, increase in moisture, and elongation, as explained in [4,6,7].

An asymmetric double lockstitch seam was used to embroider the sensor wires onto the textile backing—a modified form of stitch type '301'—with an affixing suture thread (upper thread = bobbin thread: polyester multifilament thread 'Serafil 200, yellow', or, for conductor spacing a of 0.5 mm, polyester multifilament thread 'Serafil 300/2, red', Amann & Söhne, Bönnigheim, Germany) and the electrical conductors as the lower thread (= needle thread) using a 'JF 0111-500' programmable embroidery machine (ZSK Stickmaschinen, Krefeld, Germany).

In order to systematically examine the effect of the different parameters, a range of variations of the textile sensors was manufactured, including no less than three variations for each of the expected main factors of influence.

Variations of the following parameters were made (see Figure 2 and Table 1):

d→Diameter of the electrical conductor (wire)

a→Spacing between an electrical conductor and the nearest (parallel) conductor

b→Spacing of the embroidered seams (upper thread loops) along a conductor

L, B→Length and width of the double meander sensor array

F→Area of the sensor array (F = L · B)

Table 1 shows an overview of the dressing samples produced and their structural characteristics.

**Table 1.** List of dressing samples manufactured and measured, with their characteristic structural properties.

| WD-Sample | | L · B | F | d | a | b |
|---|---|---|---|---|---|---|
| Serial | Type | /mm$^2$ | /mm$^2$ | /mm | /mm | /mm |
| D | ,d–b' 0.07–1 0.14–1 0.21–1 0.07–3 0.14–3 0.21–3 | 80 · 23 | 1840 | 0.071 0.14 0.21 0.071 0.14 0.21 | 1 | 1 3 |

**Table 1.** *Cont.*

| WD-Sample | | L · B | F | d | a | b |
|---|---|---|---|---|---|---|
| Serial | Type | /mm$^2$ | /mm$^2$ | /mm | /mm | /mm |
| S | ‚a–b' 0.5–1 | 80 · 23.5 | 1880 | | 0.5 | |
| | 1–1 | 23 | 1840 | | 1 | |
| | 1.5–1 | 22.5 | 1800 | | 1.5 | 1 |
| | 2–1 | 22 | 1760 | 0.071 | 2 | |
| | 0.5–3 | 23.5 | 1880 | | 0.5 | |
| | 1–3 | 23 | 1840 | | 1 | |
| | 1.5–3 | 22.5 | 1800 | | 1.5 | 3 |
| | 2–3 | 22 | 1760 | | 2 | |
| Fx | ‚a–B' 0.5–5 | 40 · 5.5 | 220 | | | |
| | 0.5–10 | 9.5 | 380 | | 0.5 | |
| | 0.5–15 | 15.5 | 620 | 0.071 | | 1 |
| | 1–5 | 5.5 | 220 | | | |
| | 1–10 | 9.5 | 380 | | 1 | |
| | 1–15 | 15.5 | 620 | | | |

WD sample: Wound dressing sample; L, B: Length and width of the sensor array; F: Area of the sensor array; d: Diameter of sensor wire; a: Sensor wire spacing; b: Stitching spacing.

### 2.4. Presentation of the Results

To present the results,

(1) relative water-specific changes in electrical capacitance $(\Delta C/C_0)/V_{H_2O}$ represent the ratio of the relative change in electrical capacitance $\Delta C/C_0$, i.e., $(C_1-C_0)/C_0$ (where $C_0$ is the electrical capacitance before and $C_1$ after the addition of water) to the causal added quantity of water $V_{H_2O}$.

The results are shown in graphs using

(2) trend lines, expected to show possible functional relationships, are interpolants (continuous approximation functions: 'points with interpolated lines', created with the Microsoft Excel spreadsheet program).

(3) marked error indicators of the y-coordinates are the imprecisions (random errors) calculated from the measured values with the Gaussian error propagation law of mean error.

## 3. Results

First of all, Table 2 shows the electrical capacitance $C_0$ of the dressing samples measured without the addition of water. These values are the basis for the following (relative) changes in electrical capacitance due to the standardized addition of water.

**Table 2.** Measured electrical capacities of dry dressing samples (without the addition of water).

| WD-Sample | | $C_{0,1}$ | $C_{0,2}$ | $C_{0,3}$ | $<C_0>$ |
|---|---|---|---|---|---|
| Serial | Type | /pF | /pF | /pF | /pF |
| D | 0.07–1 | 43.8 | 40.4 | 42.6 | 42.3 |
| | 0.14–1 | 26.6 | 27.5 | 26.4 | 26.8 |
| | 0.21–1 | 23.9 | - | 24.8 | 24.4 |
| | 0.07–3 | 21.4 | 21.2 | 21.0 | 21.2 |
| | 0.14–3 | 20.1 | 20.0 | 21.2 | 20.4 |
| | 0.21–3 | 21.7 | 22.2 | 21.7 | 21.9 |

**Table 2.** *Cont.*

| WD-Sample | | $C_{0,1}$ | $C_{0,2}$ | $C_{0,3}$ | $<C_0>$ |
|---|---|---|---|---|---|
| Serial | Type | /pF | /pF | /pF | /pF |
| **S** | 0.5–1 | 35.1 | 32.3 | 34.7 | 34.0 |
| | 1–1 | 22.1 | 20.5 | 20.9 | 21.2 |
| | 1.5–1 | 15.2 | 15.1 | 16.3 | 15.5 |
| | 2–1 | 11.6 | 11.3 | 10.5 | 11.1 |
| | 0.5–3 | 31.8 | 33.8 | 31.5 | 32.2 |
| | 1–3 | 17.5 | 16.7 | 17.3 | 17.2 |
| | 1.5–3 | 10.7 | 11.2 | 11.5 | 11.1 |
| | 2–3 | 8.3 | 8.5 | 8.9 | 8.6 |
| **F** | Q1–1 | 21.7 | 23.1 | 28.9 | 24.6 |
| | Q2/L2–1 | 30.0 | 44.4 | 43.2 | 39.2 |
| | Q3–1 | 61.6 | 53.7 | 51.2 | 55.5 |
| | L1–1 | 16.4 | 18.8 | 19.4 | 18.2 |
| | L3–1 | 28.2 | 38.9 | 28.5 | 31.9 |
| | Q1–3 | 14.7 | 12.7 | 11.1 | 12.8 |
| | Q2/L2–3 | 17.0 | 19.2 | 17.0 | 17.7 |
| | Q3–3 | 49.8 | 53.2 | 52.0 | 51.7 |
| | L1–3 | 16.3 | 14.5 | 15.5 | 15.4 |
| | L3–3 | 23.9 | 24.7 | 25.3 | 24.6 |
| **Fx** | 0.5–5 | 22.3 | 23.2 | 18.7 | 21.4 |
| | 0.5–10 | 25.1 | 24.7 | 23.8 | 24.5 |
| | 0.5–15 | 26.8 | 26.7 | 29.7 | 27.7 |
| | 1–5 | 13.0 | 11.4 | 12.3 | 12.2 |
| | 1–10 | 17.5 | 15.7 | 14.1 | 15.8 |
| | 1–15 | 17.3 | 18.0 | 15.1 | 16.8 |

WD sample: Wound dressing sample; $C_{0,1}$, $C_{0,2}$, $C_{0,3}$—Electrical capacitance of dry WD samples nos. 1, 2, 3 (mean of three determinations, resp.); $<C_0>$—Arithmetic mean of the electrical capacities of (usually) three WD samples.

### 3.1. Correlation with Sensor Array Area F

7.5 µL distilled water pipetted on a non-wetting surface (e.g., glass) results in a water hemisphere (a half-drop) with a diameter of about 2.5–3 mm (theoretically calculated to be 3.1 mm). After pipetting 7.5 µL distilled water onto the carrier textile (cotton jersey) used to make the wound dressing samples, a visible moisture spot also about 2.5–3 mm in diameter forms within 2 or 3 min.

To check the additivity of the changes in electrical capacitance of the dressing sensors caused by water, Figure 3 shows the ratio of the relative change in electrical capacitance to the causal change in the amount of water (referred to here as the 'water-specific change in electrical capacitance') $(\Delta C/C_0)/V_{H2O}$ across the area of the sensor array on dressing samples in wound dressing sensor series Fx (a = 0.5 mm, d = 71 µm), with stitching spacings b of 1 mm and 3 mm. Although an increase in the sensor area was expected to be accompanied by a decrease in electrical capacitance in response to the additivity of changes, this cannot be seen in either figure.

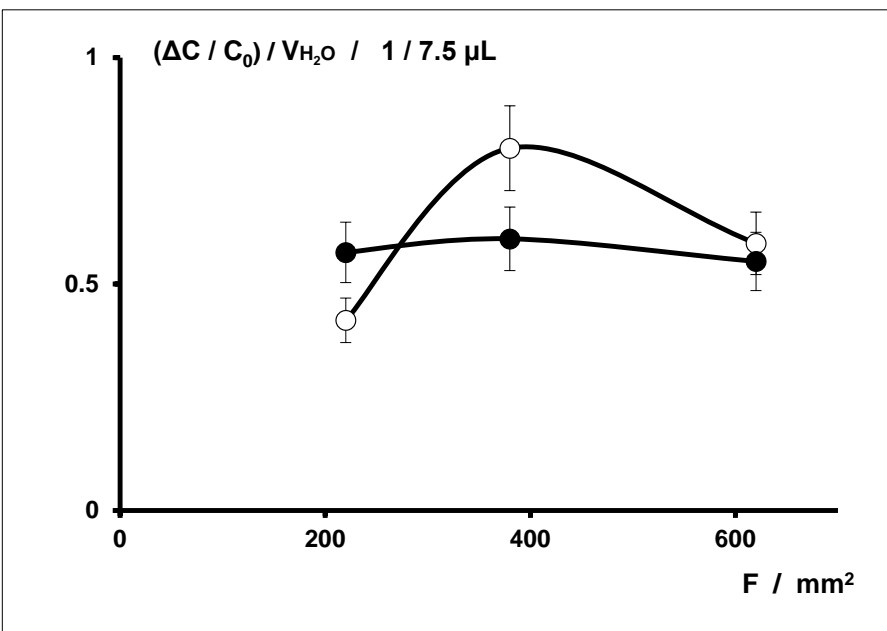

**Figure 3.** Relative water-specific change in electrical capacitance (ratio of the relative change in electrical capacitance to the causal change in water volume $(\Delta C/C_0)/V_{H_2O}$ of dressing samples in series Fx as a function of the sensor area F (= L · B). Sensor wire diameter d = 0.071 mm, stitching spacing b = 0.5 mm; Sensor wire spacing, a = 0.5 mm—black, a = 1 mm—white.

### 3.2. Correlation with Sensor Wire Diameter d

The correlation between the relative water-specific changes in electrical capacitance $(\Delta C/C_0)/V_{H_2O}$ of dressing samples from wound dressing sensor series D (L · W = 80 mm 25 mm, a = 1 mm) with stitching spacings b of 1 mm and 3 mm and the sensor wire diameter d are summarized in Figure 4.

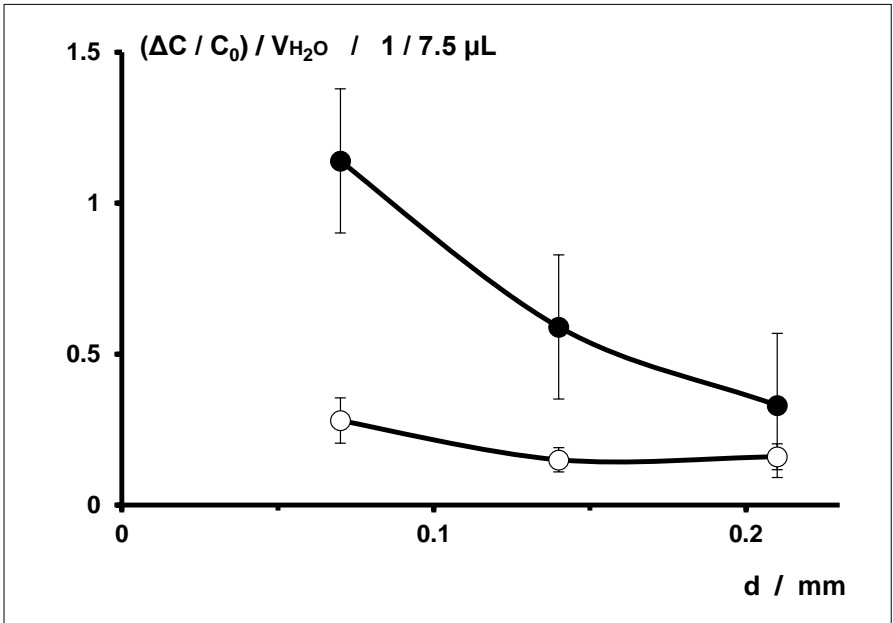

**Figure 4.** Relative water-specific change in electrical capacitance (ratio of the relative change in electrical capacitance to the causal change in water volume) $(\Delta C/C_0)/V_{H_2O}$ of dressing samples in series D as a function of the sensor wire diameter d. Sensor wire spacing a = 1 mm; Stitching spacing; b = 1 mm—black, b = 3 mm—white.

Dressing samples with stitching spacing b of 1 mm (in the range of wire diameters d studied) show a clear maximum of the relative water-specific change in electrical capacitance $(\Delta C/C_0)/V_{H_2O}$ with a sensor wire diameter d of 0.071 mm, the values declining as the diameter increases. In contrast, for dressing samples with a stitching spacing b of 3 mm, no (clear) correlation is discernible between the water-specific change in their electrical capacitance $(\Delta C/C_0)/V_{H_2O}$ and the sensor wire diameter d, the values all being approximately the same.

### 3.3. Correlation with Sensor Wire Spacing a

The correlation between the relative water-specific changes in electrical capacitance $(\Delta C/C_0)/V_{H_2O}$ of dressing samples from series S (L · W = 80 mm · 25 mm, d = 71 μm) with stitching spacings b of 1 mm and 3 mm and the sensor wire spacing a are summarized in Figure 5.

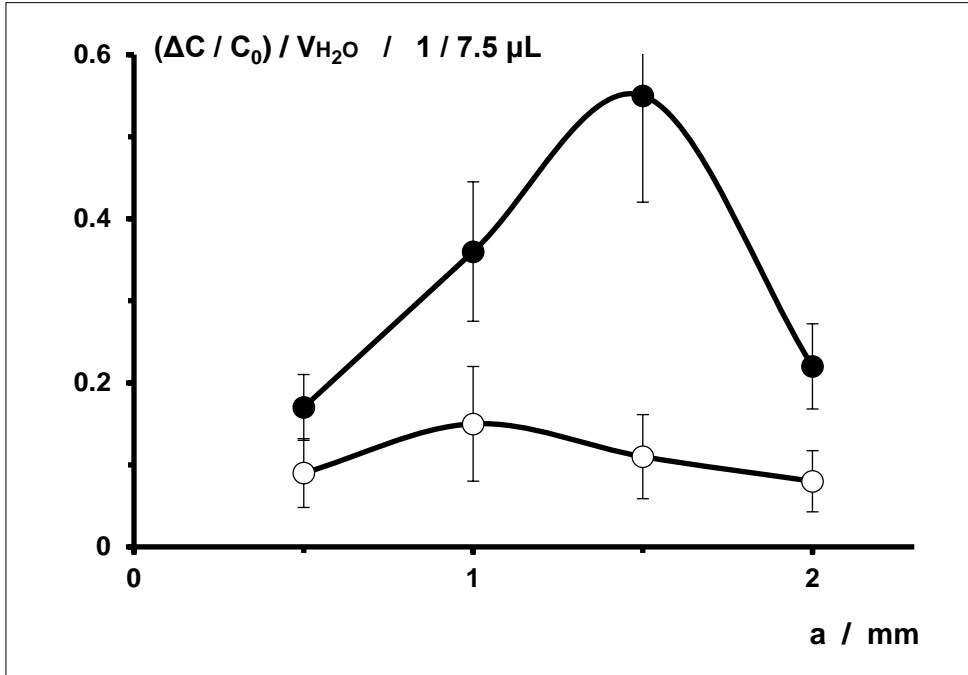

**Figure 5.** Relative water-specific change in electrical capacitance (ratio of the relative change in electrical capacitance to the causal change in water volume) $(\Delta C/C_0)/V_{H2O}$ of dressing samples in series S as a function of the sensor wire spacing a. Sensor wire diameter d = 0.071 mm; Stitching spacing; b = 1 mm—black, b = 3 mm—white.

The relative water-specific changes in electrical capacitance $(\Delta C/C_0)/V_{H2O}$ of dressing samples with stitching spacing b of 3 mm are all smaller than those of dressing samples with stitching spacing b of 1 mm (see Figure 6).

The relative water-specific change in electrical capacitance $(\Delta C/C_0)/V_{H2O}$ of dressing samples with stitching spacing b of 1 mm clearly reaches a maximum with mean sensor wire spacings a of 1 mm and, above all, 1.5 mm. For samples with stitching spacing b of 3 mm, the values are significantly lower overall; a maximum relative water-specific change in electrical capacitance $(\Delta C/C_0)/V_{H_2O}$ is indicated for a sensor wire spacing a of 1 mm.

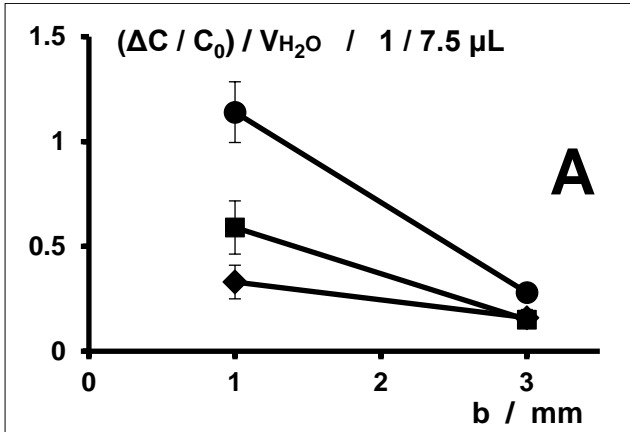 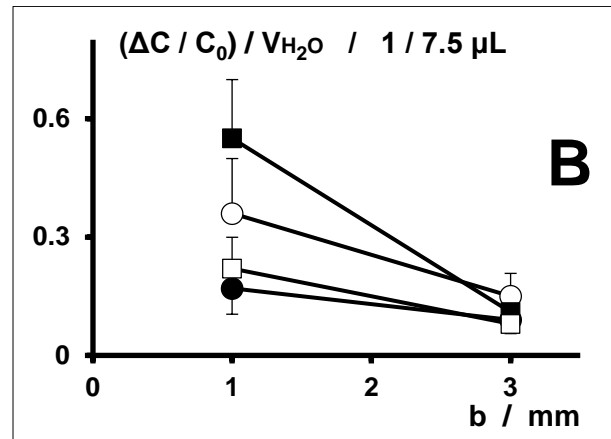

**Figure 6.** Relative water-specific change in electrical capacitance (ratio of the relative change in electrical capacitance to the causal change in water volume) $(\Delta C/C_0)/V_{H_2O}$ of dressing samples (A: series D; B: series S) as a function of the stitching spacing b (the values correspond to those in Figures 4 and 5): (**A**) Sensor wire spacing a = 1 mm; Sensor wire diameter d = 0.071 mm—•, d = 0.140 mm—■, d = 0.210 mm—♦; (**B**) Sensor wire diameter d = 0.071 mm; Sensor wire spacing a = 0.5 mm—•, a = 1 mm—○, a = 1.5 mm—■, a = 2 mm—□.

### 3.4. Correlation with Stitching Spacing b

Figure 6 shows that the relative water-specific changes in electrical capacitance $(\Delta C/C_0)/V_{H_2O}$ of dressing samples with stitching spacing b of 3 mm are all significantly lower than for samples with stitching spacing b of 1 mm.

## 4. Discussion
### 4.1. Additivity

The results regarding the additivity of changes in electrical capacitance by means of the addition of water are inconsistent. On the one hand, the correlation between the change in electrical capacitance and the amount of water pipetted is evidently additive for just one single wound dressing (cf. Figure (7) in [4]; graph of a plot of the change in electrical capacitance $\Delta C$ (= $C_1 - C_0$, the difference of the electrical capacitances after and before the addition of water) versus the volume of water added $V_{H_2O}$ (between 7.5 μL and 30 μL) was a straight line through the origin with very little scatter). For one wound dressing, therefore, the water-specific changes in electrical capacitance over the volume of water added are strictly linear, i.e., additive, when the water drops are applied to a sensor wound dressing.

When wound dressing samples with different sensor areas F are used, if the water-specific changes in electrical capacitance are additive (e.g., $(\Delta C/C_0)/V_{H_2O}$), the plot against the sensor wire area ought to show an inverse correlation. This is because the proportion of the area—and hence also the proportion of the conductive sensor structure—changing due to a standardized amount of new dielectric is reduced. However, the results (Figure 3) show that local changes in the relative permittivity in the dressing samples caused by the addition of water cannot simply be added to the unchanged parts to form a total electrical capacitance. Instead, the measurable changes are complex. One possible reason could be the uneven positioning of the sensor wires. Owing to the double meander structure, there is no directly alternating sequence of sensor wires; instead, the same sensor wire always lies next to itself twice (see Figure 2, top left). Given a moisture spot in the carrier textile of 7.5 μL water with a diameter of between 2.5 and 3 mm, this moisture may happen to be largely located between loops of the same sensor wire. Furthermore, since the same sensor wire is always at the same electrical potential, it does not form an electric field with itself. Moreover, the subjective effort to pipette 7.5 μL water not too close to the edge of the sensor array could have resulted in water being added to (around) the same sensor wire loops.

Then again, the relative changes in electrical capacitances following the very small additions of just 7.5 µL water are so large that technical implementation on an empirical basis seems easily possible.

### 4.2. Correlations with Sensor Wire Diameter d

The water-specific change in electrical capacitance (e.g., $(\Delta C/C_0)/V_{H_2O}$) showed for different stitching spacings b very different correlations with the sensor wire diameter d (Figure 4).

At a stitching spacing b of 1 mm, there is a clear correlation with the sensor wire diameter d, which grows smaller as the diameter increases. With a stitching spacing b of 3 mm, however, the correlation with the sensor wire diameter d is very low, and all the individual values for the diameters d examined are similar.

The only plausible explanation seems to be that stitching loops which are relatively close together (b = 1 mm) constrict and limit the volume of the carrier textile, and thus keep it in closer proximity to the sensor structure (steric hindrance). As a result, water flowing or diffusing into the interior of the textile increasingly enters relatively central areas of the electric fields of the electric capacitor field formed by the sensor structure, strengthening the influence of the changed dielectric (water).

### 4.3. Correlations with Sensor Wire Spacing a

The correlations between the relative water-specific change in electrical capacitance $(\Delta C/C_0)/V_{H2O}$ and the sensor wire spacing a with different stitching spacings b (Figure 5) aren't uniform, either.

With a stitching spacing b of 1 mm, correlation with the sensor wire spacing a is clear, reaching a maximum at mean spacings (a = 1 mm and a = 1.5 mm). With stitching spacing b of 3 mm, a correlation with the sensor wire spacing a is also low, the individual values at the examined spacings a being small and displaying a decreasing trend. For the relative water-specific change in electrical capacitance $(\Delta C/C_0)/V_{H_2O}$, all values are similarly small (without any apparent trend).

However, these results contradict the possible explanation put forward above of the steric hindrance of the carrier textile in the vicinity of the sensor wires. According to this explanation, the correlation between the water-specific change in electrical capacitance and the sensor wire spacing a ought to be the greatest in connection with very small spacings (a = 0.5 mm). Evidently, there is at least one other overlying effect that can also be attributed to the uneven placement of the sensor wires, especially with larger sensor wire spacings a.

### 4.4. Correlations with Stitching Spacing b

The relative water-specific change in electrical capacitance $(\Delta C/C_0)/V_{H_2O}$ showed very uniform correlations with the stitching spacing b (Figure 6). The measuring signals of comparable (i.e., otherwise identical) wound dressing samples are always higher for small stitching spacing (b = 1 mm) than for larger stitching spacing (b = 3 mm).

This in turn supports the above-mentioned explanation of steric hindrance of the area close around the sensor wires. This possible explanation is based on the assumption that more frequent stitches with shorter spacing b hold the textile backing closer to the sensor wires, enabling moisture in the textile to get closer to the sensor wires and thus between the sources and sinks of the electric fields. In addition (or alternatively), the fixing yarn used to stitch the sensor wires to the textile could become saturated with water, and thus likewise deliver more water to the immediate vicinity of the conductors and keep it there. Both mechanisms would coherently explain why the changes in electrical capacitance are greater at a stitching spacing b of 1 mm than at 3 mm.

## 5. Conclusions

This systematic investigation of textile sensors for the detection of moisture, consisting of insulated sensor wires stitched onto a textile backing, shows that the measurements

correspond in principle to the preceding theoretical considerations (even if the effects found can't be explained with absolute certainty) and that the magnitude of the sensor signals is probably well sufficient for the purpose stated. The investigation shows that there are probably no insurmountable technical obstacles preventing the development of purely textile sensor wound dressings which can detect water. Enamel-insulated electrical conductors can be stitched onto textiles. Double meanders provide a suitable geometrical structure which allows these wires to act as sensors for electrical capacitance. Changes in capacitance enable the condition of primary healing wounds (medically treated wounds, i.e., which are sutured or stapled) to be monitored by revealing postoperative wound healing disorders in the form of bleeding or discharge from a seroma.

As the main conclusion, the design of a final sensor wound pad combining the measurement of temperature, fluid, and stretching regarding the measurement of fluid can be planned based on a set of functional dependencies as determined and described in this manuscript.

**Author Contributions:** Resources, project administration, funding acquisition, and supervision: K.Z.; Investigation: H.P.; Conceptualization, methodology, validation, formal analysis, writing (original draft preparation, and editing), and visualization: H.P. and K.Z. Authorship was strictly limited to those who have contributed substantially to the work reported. All authors have read and agreed to the published version of the manuscript.

**Funding:** This research was funded by Deutsche Forschungsgemeinschaft (DFG) grant number ZI 1518/3-1.

**Institutional Review Board Statement:** Not applicable.

**Informed Consent Statement:** Not applicable.

**Data Availability Statement:** Data sharing is not applicable, all relevant data are contained in the article.

**Conflicts of Interest:** The authors declare no conflict of interest.

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
