# Peer review of "Monitoring of Surgical Wounds with Purely Textile, Measuring Wound Pads—III: Detection of Bleeding or Seroma Discharge by the Measurement of Wound Weeping"

_textiles, doi:10.3390/textiles2040031_

Round 1

Reviewer 1 Report

As per attachment

Reviewer 2 Report

A great overview of the possibilities for measuring humidity with the embroidered sensors. However, there are very few measuring points for a fit in the diagrams. Especially in Figure 6. But this probably does not mean linear dependence?

In Figure 3, the curve with three measuring points is also very hard to fit. Are there some more points available?

Reviewer 3 Report

In this manuscript, authors investigate the textile sensors for the detection of wound weeping. After I read the article, I think the overall logic of the article is confusing, while some basic presentation is not standardized, and the author needs to redesign the whole manuscript from scratch, while the writing style needs to be standardized so as to increase readability. I look forward to seeing the revised version before making a decision

1.      The abstract does not follow the general logic, while too many paragraphs make it difficult to figure out what the author's conclusion is.

2.      In the introduction section, the author asks what questions are needed to be addressed in this article?

3.      What is the author's main conclusion, I did not get an answer?

Reviewer 4 Report

The paper entitled “Monitoring of Surgical Wounds with Purely Textile, Measuring Wound Pads – III. Detection of Bleeding or Seroma Discharge by the Measurement of Wound Weeping” focuses on the identification and quantification of possible variables affecting the detection of water with purely textile, measuring wound pads in a systematic study to technically optimize the dressing-integrated sensors. The tests used are not versatile but they are enough to fulfill the purpose set out by the authors. Although it can be improved, the introduction refers to the aim of the study. The experimental part is consistently revealed and explained while the results and discussions are understandably submitted. The conclusion summarizes the aforementioned results. In my opinion, the paper may be interesting from a scientific and practical point of view.

However, I would like to recommend the publication of the manuscript in this journal after fulfilling the following recommendations:   

1.     The abstract could be shortened while keeping the qualitative findings with enough

2.     precision;

3.     The introduction part should be more focused and address the issues raced in the experimental part of the manuscript;

4.     While measuring the spacing between electrical conductors or spacing of the embroidered seams along a conductor, do the authors envisage the patient’s movements that can change these spacings?

5.     Do the sensors wires make the textile for wound dressing non-flexible, rough and inconvenient for squeezing to the wound?

6.     The statements that “wires made of silver” are “non-toxic” while “copper is toxic to cells (and thus inhibits healing)” are not true. Ag is a moderately toxic and non-essential element for humans. Cu is an essential trace element for humans, with an average daily dietary requirement of 2 mg. Furthermore, copper can be metabolized [DOI: 10.1002/jbm.b.31412], whereas silver tends to resist metabolization, increasing body’s silver serum level [doi: 10.1002/1097-4636(200009)53:5<600::aid-jbm21>3.0.co;2-d.]. Except for its antibacterial activity, Cu has also a potential angiogenic effect [doi: 10.1089/ten.tea.2009.0504.].

7.     It would be better for the authors to combine section 3 (Results) and section 4 (Discussion) for easier interpretation of the results.

Round 2

Reviewer 1 Report

Acceptable

Author Response

There is no response necessary, thank you.

Reviewer 3 Report

1、  What is the benefit of these textiles compared to other molecules in the sensing application?  some articles can be cited and compare, Advanced functional materials, 2020, 30(2): 1902634. Biomater. Sci., 2022, DOI: 10.1039/D2BM00719C ; Journal of Controlled Release, 2022, 349: 963-982.

Author Response

Authors Reply:

The textile-based sensors (without the involvement of ‘new’ sensoring molecules)

- measure purely physical,

- are able to measure various quan­tities (virtually) simultane­ously, and

- have no capability of a controlled drug release.

Reviewer 4 Report

 Although the changes in the manuscript were minor, it paper may be published in its present form.

Author Response

There is no response necessary,

thanks to the reviewer.